# Using Data Integration to Improve Health and Welfare Insights

**DOI:** 10.3390/ijerph19020836

**Published:** 2022-01-12

**Authors:** Linda R. Jensen

**Affiliations:** Australian Institute of Health and Welfare, 1 Thynne St., Bruce, ACT 2617, Australia; linda.jensen@aihw.gov.au

**Keywords:** continuity of care, data, disability, dementia, health, health service use, integration, last year of life, linkage, suicide, veterans, welfare, wellbeing

## Abstract

The Australian Institute of Health and Welfare (AIHW) is a leader in the provision of high-quality health and welfare information. Its work program has built a strong evidence base for better decisions that deliver improved health and welfare outcomes. The evolution of the AIHW’s data integration program has exemplified innovation in identifying and addressing key information gaps, as well as responsiveness to opportunities to develop and capture the data required to inform national priorities. The AIHW conducts data integration in partnership with data custodians and specialists in integration and analysis. A linkage project requiring the integration of Australian government data must be undertaken by an accredited integrating authority. The AIHW has met stringent criteria covering project governance, capability, and data management to gain this accreditation. In this capacity, the AIHW is trusted to integrate Australian government data for high-risk research projects. To date, the AIHW’s integration projects have generated improved research outcomes that have identified vulnerable population groups, improved the understanding of health risk factors, and contributed to the development of targeted interventions. These projects have fostered new insights into dementia, disability, health service use, patient experiences of healthcare, and suicide. Upcoming projects aim to further the understanding of interrelationships between determinants of wellbeing.

## 1. Introduction

The Australian Institute of Health and Welfare (AIHW) is a leader in health and welfare information. Its analytical work program offers insights into how Australians interact with the health and welfare systems, through the analysis of a broad range of data sourced from surveys, administrative records, and service delivery functions. The outputs of the AIHW work program provide strong evidence as a basis for better policy and program delivery decisions that ultimately lead to improved health and welfare outcomes for Australians [1].

The AIHW is an international leader in data integration and has gained accreditation as an integrating authority [2,3]. This accreditation allows the AIHW to integrate Australian government data for statistical and research purposes.

In this context, the AIHW has developed an extensive data integration program that has provided opportunities to build richer analytical datasets, which can deliver research outcomes beyond what is possible from analysis of single data sources. The program of work has provided opportunities for the following:Broader level reporting and analysis—e.g., whole-population and national data;Addressing key data gaps by connecting content datasets that relate to a single entity;Analysis and reporting of rare or sensitive issues and events;Analysis of pathways taken through health and welfare systems and to understand the experience over a person’s life course;Identification of specific population groups in broader administrative datasets—e.g., migrants or veterans.

The AIHW adheres to all relevant legislation and guidelines to ensure it upholds strict privacy and confidentiality requirements.

The AIHW seeks opportunities to drive innovation in the collection, use, and analysis of health and welfare information, to learn from international experience, and to develop approaches that align with best practice. The AIHW’s international engagement includes the following:Participation in Organization for Economic Co-operation and Development (OECD) activities, including the Working Party on Health Statistics;Longstanding involvement with the World Health Organization, through the WHO Family of International Classifications Network (WHO-FIC), and as the designated Australian Collaborating Center (ACC) for the WHO-FIC;Collaboration with the Canadian Institute for Health Information, through sharing and comparing approaches, and participating in secondments across the agencies;Membership with the National Initiative Network, a collective that shares experiences in developing stronger frameworks to promote secondary use of health and wellbeing data;Partnership with The Five Eyes research collective, with a particular focus on international comparisons of data about veterans;Contact with the United States of America National Center for Health Statistics, Statistics New Zealand, and the Commonwealth Fund.

## 2. AIHW Data Integration Projects

To date, data integration projects undertaken by the AIHW have generated improved research outcomes for a number of specific population groups and health and welfare topics. This has supported the identification of vulnerable population groups, provided a better understanding of health and welfare risk factors, and contributed to the development of targeted interventions. The AIHW has addressed significant data gaps, extended analysis of complex relationships in health and welfare, and undertaken pioneering research that provides new insights into how Australians interact with health and welfare systems.

Projects are currently underway which seek to maximize the use of broad, national data assets to build a more holistic understanding of the interrelationships between the determinants of health and wellbeing for Australians. Some of these assets are held and maintained by the AIHW, while others are accessed in partnership with other Australian government agencies.

All AIHW data integration projects are conducted in partnership with a range of data custodians and specialists in data integration and analysis. Recent and upcoming AIHW data integration projects are described below.

### 2.1. Dementia

Dementia is a condition that is not consistently captured in individual datasets and is generally poorly captured in nationally representative surveys. The AIHW’s capacity to monitor and report on dementia has been transformed through data integration, which has enabled the following key achievements:Better identification of people with dementia in Australia, leading to better coverage in reporting and more accurate understanding of disease prevalence, comorbidities, risk factors, and population groups with dementia.Developing an understanding of the course of disease over time for people with records of dementia including the potential to examine factors that affect the use of health and aged care services. Records of dementia may include a specific diagnosis, or recorded use of dementia-specific medications for diagnosis by proxy.Understanding the consequences of dementia diagnoses in many more aspects of a person’s life than previously reportable—e.g., on work and income, or receipt of welfare or disability support payments.

Data integration has also extended the use of existing data. For example, the AIHW has developed models that identify predictors of early dementia in a dataset that contains no dementia diagnosis information [4]. Ultimately, this information can be used to more accurately estimate dementia incidence and prevalence, as well as contribute to filling information gaps in the primary and specialist care domains.

The AIHW’s dementia data integration projects are providing new and important information on people with dementia in Australia, making use of data from the five-yearly Australian Census of Population and Housing, plus welfare data that would previously have been of little use for studies of dementia without data integration.

### 2.2. Disability

The AIHW is working in partnership with national and state-level governments to integrate government data to develop a National Disability Data Asset (NDDA). This project brings together deidentified data from over 50 datasets, sourced from all levels of government, to build a linked administrative data asset that can support reporting under the outcomes framework of a new National Disability Strategy (NDS).

The NDDA is currently in its pilot phase, which is focused on developing processes for sharing data among government data custodians, to gain a better understanding of people’s life experiences. Analyses of five public policy topics are being used to demonstrate the potential of using linked data, as well as to inform design for a potential enduring asset. These topics include early childhood support, experiences with the justice system, pathways from education to employment, and services and support for people with disability and mental health issues. The pilot phase aims to derive a comprehensive measure of disability and demonstrate how linked administrative data can support an outcomes framework under the NDS.

This pilot builds on AIHW’s health and welfare data expertise, which has supported NDDA delivery partners to develop and capture previously unavailable information on people with disability. The complexity of negotiating ethical approval, navigating Australian government legislation requirements, and ensuring privacy compliance of personal information has proven challenging. As the integrating authority for the pilot phase of the NDDA, the AIHW has established rigorous end-to-end data governance and management arrangements in accordance with privacy, legal, and technical aspects of the supply, to ensure that data of value can safely be included in the asset. A key achievement is the collaboration among the AIHW, the Australian Bureau of Statistics (ABS), and state government partners to create the pilot dataset.

Early learnings from the pilot include the potential to improve data sharing arrangements by streamlining governance and leveraging existing data integration infrastructure at a national level. A system that facilitates delivery of timely and relevant data will inform national priorities and support improved policy development, program design, and service delivery for people with disability.

Research findings on the five topic areas will be available in late 2021. Learnings from the pilot will inform options for an enduring data asset beyond 2021, including priority data for inclusion, data integration models, approved uses of the NDDA, and appropriate governance models for the asset.

### 2.3. Health Service Use: Last Year of Life

The AIHW is using the National Integrated Health Services Information (NIHSI) integrated data asset to examine health service use patterns and their corresponding costs for Australians who lived their last year of life between 2011–2012 and 2016–2017. The project aims to identify key factors related to the variability in the patterns of health service use in the last year of life. The key factors may include patient characteristics of age, sex, remoteness, socioeconomic group, and cause of death. Comparisons will be made to the health service use of the rest of the population (those who did not die) with otherwise similar characteristics.

Key analysis datasets used in this project are the National Death Index (NDI) linked to data on health service use, pharmaceutical prescriptions, and hospital, emergency department, and residential aged care.

Results from this analysis will provide information on Australians’ interaction with a range of health services prior to death. They will help to identify the characteristics of Australians who are not accessing the services they need in their last year of life. This will provide useful information for healthcare professionals and policymakers.

Analyses for both service utilization and costs are underway, with an interactive web report planned for release in late 2021.

### 2.4. Patient Experiences of Continuity of Care

The AIHW developed the Coordination of Healthcare (CHC) study in partnership with the ABS to fill an important data gap and provide information on patients’ experiences of continuity of care across Australia [5]. The study, which included people aged 45 and over, used a survey and data integration to examine patient experiences of continuity of care across Australia and importantly by the Primary Health Network [6]. The survey collected self-reported experiences of health service use including general practitioners (GPs), specialists, hospitals, and emergency departments. It also asked about health status including long-term health conditions, medication use, and sociodemographic characteristics.

Responses from consenting participants were linked to their administrative health data, including health service use and pharmaceutical prescription data, plus hospital and emergency department data for the pre- and post-survey period. The resulting integrated dataset provides a unique source of information on patient experiences, health status, and service use data [7]. This data linkage has enabled researchers to quantify and describe the actual use of health services (such as GP and hospital visits) and compare it with self-reported data on people’s experiences of healthcare.

### 2.5. Suicide

The AIHW has used data from the Multi Agency Data Integration Project (MADIP) national data asset to analyze the contribution of different social determinants to death by suicide in Australia. The results will inform future policy development to help prevent deaths by suicide.

The 2011 Australian Census of Population and Housing was linked to the ABS Death Registrations collection to form the analysis population, which was then linked to key analysis datasets, covering a range of health and welfare topics. The quality of income data from MADIP was improved using a synthetic measure developed by the Australian National University, based on taxation and social security payment data. Application of a weighting methodology addressed issues with linkage coverage.

Initial analyses on cumulative risks of dying from suicide by educational attainment and employment status were released publicly [8].

Multiple statistical models have been developed (including a competing risk model) to provide a better understanding of the contribution of different social determinants—sex, age, indigenous status, occupation, marital status, household composition, and personal income—to deaths by suicide. Preliminary results, intended for publication in September 2021, provide useful insights into associations between certain social determinants and the risk of suicide.

### 2.6. Veterans

Because of their unique service experience, many permanent, reserve, and ex-serving Australian Defense Force (ADF) members (‘veterans’) and their families experience challenges beyond those typically experienced by the general Australian population [9,10].

In collaboration with the Australian national government’s Departments of Defense and Veterans’ Affairs, the AIHW is monitoring and reporting on the health and welfare status of veterans. Outcomes for veterans are compared to those of the broader Australian population, to identify specific risk and protective factors, as well as the social determinants of health and welfare of Australian veterans.

To date, specific analyses in this work program have focused on understanding the following:Overall causes of death and incidence of suicide for current serving and ex-serving ADF members [11];The welfare of ex-serving ADF members, from analysis of several topics including housing, social support, education and skills, employment, and income and finance;The use of healthcare services by ex-serving ADF members;Use of subsidized prescription medication by ex-serving ADF members;Health status, risk factors, and health conditions [12].

Under the veterans’ analysis work program, the AIHW integrates ADF personnel data with other government datasets to enable identification of veterans in a range of administrative datasets including death registries, Pharmaceutical Benefits Scheme, and Homelessness Support Services. A key development resulting from this work program has been the addition of a veteran identifier flag to the MADIP, which will allow identification of veterans in the broad range of datasets included in MADIP, to inform on aspects such as employment, income, education, and social service use.

The analytical outputs of the veterans’ analysis work program are building a profile of Australia’s veterans which is helping to drive data improvement, inform development of policy and targeted interventions, and ultimately, improve the wellbeing of Australia’s veterans [13].

## 3. The AIHW Data Integration Process

Australian government entities strongly support data integration, to maximize the benefits and use of government data assets and, importantly, to reduce the burden on individual respondents and data providers.

To protect privacy and confidentiality, as well as maximize the public benefit of its research, the AIHW integration program takes place in a secure and regulated environment. The AIHW integration environment is characterized by strict adherence to privacy principles through appropriate governance and approvals, strong strategic partnerships, secure data integration processes, and the creation and maintenance of high-quality data assets.

### 3.1. Governance and Approvals

The AIHW’s data integration environment adheres to, and is bound by, both mandatory requirements and best practice policies and processes. These include Australian government legislation, policy, and guidelines, data security protocols, approval by ethics or human research ethics committees, data custodians and data access committees, and adherence to national and international best practice and frameworks including the *Privacy Act 1988* and the *Australian Privacy Principles (APPs)* [14,15]. The AIHW’s data governance framework provides details of our strong data governance arrangements, including descriptions of key concepts, governance structures and roles, and the systems and tools that support them [16]. In addition, the AIHW’s privacy policy outlines how the AIHW handles personal information [17].

The AIHW Ethics Committee is established under Section 16 of the *Australian Institute of Health and Welfare Act* [18]. Its functions and membership are prescribed in the *Australian Institute of Health and Welfare (Ethics Committee) Regulations 2018* [19]. All data linkage projects must be approved by the AIHW Ethics Committee and other relevant ethics committees where appropriate. The ethics application must include evidence of consultation with relevant stakeholders, including the general community, to establish their support and trust. Projects are required to be transparent and must make results publicly available. As part of this, information about projects and their outcomes are also published on the AIHW website.

All projects are assessed against the Five Safes, which is an internationally recognized approach to considering strategic, privacy, security, ethical, and operational risks as part of a holistic assessment of the risks associated with data sharing or release [20]. Guided by this framework, the AIHW applies the following criteria to assess all new integration projects and assign a risk rating:Safe Projects—Is the use of the data appropriate (legal, moral, and ethical)?Safe Users—Can the users be trusted to use it in an appropriate manner?Safe Data—Is there a disclosure risk in releasing the data itself?Safe Settings—Does the access facility prevent unauthorized use?Safe Output—Are the statistical results non-disclosive [21]?

Output is governed by the *Australian Institute of Health and Welfare Act 1987,* and strict review of outputs by AIHW’s data integration managers ensures protection of privacy and confidentiality [18].

### 3.2. Strategic Partnerships

Strong relationships with stakeholders are essential to the production of accurate information and to achieving improved data collection practices. The AIHW collaborates closely with experts in data integration and has effective data partnerships with government entities—national, state, and local, as well as with universities, research centers, nongovernment organizations, and other experts throughout the country. For current integration projects, the AIHW engages with government experts on health, education, community services, and housing.

The AIHW is working closely with the ABS, to develop a consistent national integration system. This partnership aims to maximize the use of existing survey, Australian Census of Population and Housing, and government administrative data. By supporting the use of consistent national data standards and approaches to collection, the AIHW and ABS can build and coordinate secure access to integrated national assets that support multiple analytical uses.

All data integration work is performed under guidance from a number of specialist advisory committees. In addition, the AIHW engages with consumers to continue to build community trust.

### 3.3. Quality Data

The AIHW collects, hosts, analyzes, and disseminates data that support the understanding of important health and welfare issues, and that are critical to good policymaking and effective service delivery.

The AIHW Quality Management Framework (QMF) is used to manage risk and maintain quality. The QMF draws on aspects of separate enterprise architecture, quality gate, data validation, and project management models developed by other national and international organizations. Application of the QMF across all stages of data integration and analysis projects maximizes the potential to deliver outputs that support and inform policy development and decision making.

The five key elements of the QMF are as follows:Statistical risk—Managing statistical risks, which can occur at all stages and levels in the statistical production cycle, is key to maintaining data quality. To minimize statistical risk, the QMF provides clear definitions of the risks to data quality, as well as their significance (major, medium or minor), and provides guidance on developing strategies for their management.Project management—All statistical projects must complete a risk assessment at the planning stage. The project brief lists major risks, and any risks already realized are elevated to issues. Strategies for mitigation and management must be included. The risk assessment feeds into the design of the quality assurance and data validation strategies for each project. Risks are reviewed regularly throughout the project’s lifecycle.Quality assurance (QA)—QA strategies are particularly useful for identifying medium-level statistical risks and quality issues. They give a more detailed view of the factors impacting risks and data quality, often from a process perspective. The QMF provides context, generic tools, and a broad operational model to assist with the design of consistent QA strategies. It uses a set of generic gates to improve the early detection of errors or flaws in production processes. It also defines the roles and responsibilities for managing quality and performance measures to facilitate quality gate assessments.Data validation—Data validation processes present the last opportunity to detect, resolve, and treat important errors before the data are released to clients. Validation also enables anomalous data that are correct to be identified and explained. The QMF provides templates, guidance, and explanatory notes to assist with data validation work.Reference models—The QMF is based on reference models that integrate critical project management activities into the statistical production process, provide guidance on quality assurance and data validation work, and define the roles and responsibilities of stakeholders across project phases. These models can also be used to benchmark, monitor, understand, and streamline production processes, improving responsiveness and capability into the future.

### 3.4. Data Integration

The AIHW provides a secure linkage environment for all approved linkage activity. The Data Integration Service Center (DISC) is a separate computer network that is not connected to the internet or any other AIHW system and includes strict protocols and procedures for physical security, data security, and manager review of outputs to ensure ethics compliance.

All AIHW data integration activities, regardless of risk level, are undertaken within the DISC. The linkage process is designed for each new project, around the following principles:The separation principle means that no one working with the data can view both the linking (identifying) information (such as name, address, date of birth) together with the merged analysis (content) data (such as clinical information, health service, or medication usage) in an integrated dataset. Under the separation principle, data integration is performed in three stages—separation, linkage, and merging. Each stage has a separate domain within the specific project in the DISC. Each domain is accessible only by staff holding the specified role, and staff members can only perform one role in each project.Linkage is done on datasets containing essential data items only,Sophisticated probabilistic data linkage methodology is used to achieve the best possible linkage results. The linkage is performed using linkage software developed by the AIHW.Output is appropriately confidentialized before it is made available to researchers, in accordance with appropriate legislation and the requirements of data custodians.

## 4. Opportunities and Challenges

Data integration has increased the capacity to fill key data gaps and support better decisions to develop and deliver targeted interventions to those who are at risk. By combining data from different sources, and harnessing expertise through strategic partnerships, the AIHW data integration program has provided opportunities for the following:Enhanced analysis—the research potential of integrated datasets is greater than of those based on a singular source. Integrated data have a broader coverage of topics and provide greater potential to examine interrelationships between topics.Cleaner data—the combination of data from different sources enables the development of improved data checks that can enhance the quality of the separate data sources. This can be achieved through the development of data collection standards or definitions of data items that relate to standard classifications.Cost effectiveness—linking data collected for other purposes is far cheaper than obtaining similar data through surveys and longitudinal studies. Reuse of existing administrative data greatly reduces the costs associated with both provision and collection.Improved coverage—linked datasets can represent a large sample, allowing broader-level reporting that is not possible from individual survey data. Use of integrated data can assist to address issues associated with small numbers. This both protects the privacy of individuals and enhances analysis and reporting of sensitive or rare events.Identification of target groups—linking information about target groups, e.g., migrants or veterans, creates broader datasets that support the analysis of wellbeing and identification of risk factors for these groups, without the requirement to ask for this detailed information in administrative datasets.Longitudinal analysis—integrating datasets over time can allow the analysis of pathways through health and welfare systems or over the life course of an individual or cohort.

While offering many opportunities, the development of and access to a broad range of data assets for integration present many challenges such as the following:Coordination of the large and complex data integration landscape, involving data assets from all levels of government.Complex governance arrangements—including understanding the implications of relevant legislation, policies, and ethics. This requires a considerable amount of time and documentation.Managing liaison and approvals across multiple data custodians, especially across different levels of government. Integration projects drawing data from multiple sources typically require approvals from multiple ethics committees or custodians.Building community trust and engagement.Methodological challenges, where weighting practices may be required to ensure appropriate representativeness of the data.Data inconsistencies across input data sources. Data used in integration projects are often collected for service delivery or administrative purposes. They may have different definitions, concepts, specifications, coding, classifications, standards, and quality across sources.The need to quickly develop expertise in new and complex data models, as well as new approaches to analysis. As demand continues to grow for accessible and large-scale linked data assets such as the NIHSI and NDDA, the AIHW is responding to more complex, cross-sector research questions.

In many cases, the source data for AIHW integration projects have not been used in this way before. The AIHW has invested considerable effort to assess their suitability to inform policies and meet research objectives through data integration.

To meet these challenges, maintain strong leadership in data integration, and ensure the ongoing utility of Australian government data to meet research objectives and inform decision making, the AIHW data integration work program will continue to focus on the following:Forming new and strengthening existing partnerships across all levels of government, to promote access to and use of data assets, as well as sharing of expertise.Promoting processes to safely share data for integration in national data assets that allows richer, deeper analysis of populations of interest. An example of this is the addition of population flags to national data assets, as in the AIHW veterans’ analysis work program.Continuous improvement and innovation in data collection practices, including opportunities to harmonize the way data on topics of interest are defined and collected.Supporting the development of governance frameworks that facilitate data integration involving data assets from all levels of government, while maintaining the privacy and confidentiality of data about individuals, as well as meeting the specific requirements of data custodians for access to and use of their data.

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
