# Peer review of "Using Data Integration to Improve Health and Welfare Insights"

_ijerph, 2022, doi:10.3390/ijerph19020836_

Round 1

Reviewer 1 Report

Dear Authors,

thank you very much for writing this article, I've learned a lot of new things.

Three minors comments:

1) One thing I would have liked to read is how the AIHW fits in the international context. Are there similar agencies around the world? If so where and do they work like the AIHW?

2) I liked the pilot project section but are there finalized project that have already produced sono policy o research relevant outcomes?

3) Please check the font at page 4

Author Response

Thank you so much for your encouraging review. I have addressed your comments and revised the article accordingly. Please see the attached file for specific responses.

Reviewer 2 Report

This paper attempts to introduce a data integration program conducted by the Australian Institute of Health and Welfare (AIHW). Few specific AIHW data integration projects and the integration process are presented. Opportunities and challenges of AIHW are also mentioned. The AIHW data integration program is indeed a useful and meaningful. However, this paper does not like an academic article.

  1. The paper does not demonstrate an adequate understanding of the relevant literature in the field and cite an appropriate range of literature sources. The paper should have a clear research question and a thorough review of related literature.
  2. The paper's argument is not built on an appropriate base of theory or concepts. This paper should employ a theoretical lens to answer the research question.
  3. There is little method employed. For example, a qualitative method can be used to analyse second hand materials to find out the key elements during data integration. Alternatively, a design science method can be in the design of the data integration process.
  4. Results are not clearly presented or appropriately analysed?
  5. The paper does not clearly identify any implications for research or the gap between theory and practice.

Author Response

Thank you so much for your review of this article. I have responded to your comments - please see the attachment.

Round 2

Reviewer 2 Report

If this manuscript’s submission type is “Article", all my comments on the previous version remain unanswered, except that the author has strengthened the reference. Having read the author’s reply, I think this manuscript should be submitted as another proper type of submission rather than “Article”.

Author Response

The article type has been changed to 'communication'. Please advise if any further revisions are required.

This manuscript is a resubmission of an earlier submission. The following is a list of the peer review reports and author responses from that submission.